# Factors Affecting Fintech Adoption: A Systematic Literature Review

Egi Arvian Firmansyah *, Masairol Masri, Muhammad Anshari and Mohd Hairul Azrin Besar

School of Business and Economics, Universiti Brunei Darussalam, Jalan Tungku Link,
Seri Begawan BE1410, Brunei
* Correspondence: 21h1102@ubd.edu.bn

**Abstract:** The rise of financial technology (fintech) has been one of the substantial changes in the financial landscape driven by technological advancements and the global financial crisis. This paper employs the systematic literature review (SLR) technique to review recent literature on fintech adoption or acceptance employing the Scopus database (2019–2022). The final reviewed documents are sixteen journal articles published by various journals from different country contexts and theoretical backgrounds. Several inclusion criteria were used to filter those selected documents. One crucial criterion is the journal continuity in the Scopus index, which assures the quality of the published scholarly works. This criterion selection is expected to represent this paper's novelty. The study reveals various determinants derived from the theories used by the fintech researchers. However, the Technology Acceptance Model (TAM) and the Unified Theory of Acceptance and Use of Technology (UTAUT) are the most used theoretical foundations. Additionally, trust, financial literacy, and safety are other factors developed by previous researchers and are significant determinants of fintech adoption. Besides, these results suggest that future studies on fintech adoption develop a genuine construct since fintech keeps progressing, and so does the customers' behavior.

**Keywords:** adoption; fintech; review; systematic literature review





## 1. Introduction

Financial technology (fintech) can be defined as technology application in the provision of various financial services [1]. Fintech companies are innovative financial intermediaries that employ technological advancements to support novel business models, adjustments to operational procedures, and the provision of enhanced goods and services [2]. Fintech first appeared in the early 1990s, along with the internet revolution. The Internet has been one of the pivotal factors determining the growth of the fintech sector [3]. However, the fintech study, primarily available in the Scopus database, was first documented in an article by Mackenzie in 2015 [4].

Fintech is believed to offer the chance to make finance more transparent, consumer-friendly, and cost-effective. Besides, it has also been considered to revolutionize the financial landscape by challenging the incumbent financial service providers, such as banking, insurance, and existing investment companies. In addition to technological advancement, fintech proliferates because it possesses dissimilar regulatory force from the existing financial service providers, allowing fintech firms to operate more flexibly under the regulatory sandbox to create innovative products [5].

The fintech ecosystem consists of several elements with various business models and services [6]. The fintech ecosystem comprises fintech startups, technology developers, the government, customers, and traditional or existing financial institutions. Furthermore, the fintech business model may include payment, wealth management, crowdfunding, peer-to-peer (P2P) lending, capital market, and insurance (insurtech) business models [7].

In line with the practical side, fintech studies have been growing in the last few years, coming from different fields or focuses. Besides, review papers on fintech are also apparent.

For instance, a review study by [8] reveals that most fintech studies come from Asia and the European Union and employ case-study research methods. Meanwhile, Ref. [9] performs a systematic review of fintech in relation to Islamic finance while providing future direction. Furthermore, Ref. [10] presents almost similar results and setting but using a slightly different technique, namely a hybrid approach by mixing bibliometric and content analysis to reveal the current research trend of Islamic financial technology. Still related to fintech and the Islamic setting, a study by Ref. [11] focuses on Islamic fintech for SMEs and digitalization readiness. One research closely related to this current study is a previous study by Ref. [12]. However, some differences between the two. First, this present study employs an explicit keyword of 'fintech adoption' in the search bar, while Ref. [12] mentions more terms related to fintech, such as 'product adoption', 'online lending', and 'financial technology'. In other words, the used research strings are dissimilar. Second, this present study uses a single yet extensive database, namely Scopus, while Ref. [12] uses six smaller and more specific databases, such as Emerald, Sage, and Wiley. Third, the produced conceptual framework in this present study focuses on theoretical foundations underpinning fintech adoption factors, while Ref. [12] highlights the standpoints of fintech innovators and adopters. Articles using the keywords 'adapt, adaptability, or adaptation' are also excluded from this study because adoption and adaptability have different meanings e.g., Ref. [13].

Pertaining to the growing fintech ecosystem, business models, and practices, understanding the factors which determine the acceptance or adoption of fintech services in literature remains essential, primarily to evaluate and map the motivation of customers choosing fintech services. As shown previously, this present study employs a different approach in reviewing the literature on the determining factors of fintech adoption. The result of this paper is expected to add to the literature of fintech and systematic review study by offering an explicit topic related to one branch of finance literature. Finally, this paper proposes a conceptual framework to understand fintech adoption determinants available in scholarly literature.

The purpose of this research is to review the academic literature on the factors determining the usage or adoption of financial technology (fintech) services among customers, not others, such as financial institutions or small businesses see, for example, Ref. [14]. This study treats adoption intention and actual usage as similar things in the literature because both may indicate exposure to adoption. The literature studied in this research is the published documents in various journals indexed in the Scopus database. Using the protocol outlined in the method section, we obtained sixteen final documents (journal articles) stating the keywords 'fintech' and 'adoption' in their title fields and categorized them into business and economics subject areas. Those documents are relevant to the purpose of this study, namely, investigating the factors which determine fintech service usage or adoption among fintech users or customers in various countries. Books, editorials, and book chapters are excluded from this study because we focus only on peer-reviewed documents, namely journal articles. Further filtering also automatically omitted conference proceedings because we found no document in this category.

Furthermore, this study also performs further filtering for two reasons. First, the grey papers were removed from the eligible documents because even though they mention the word 'adoption' and 'fintech', they do not deal with customer adoption. Five papers were found to be included in this category. Second, this study only includes the journals still indexed in Scopus, meaning any delisted or discontinued journal from Scopus is not considered due to quality reasons. Based on our review, two journals are excluded because they are no longer indexed in Scopus.

Journal continuity adopted in the inclusion criteria in this study is one factor that makes this current study unique because many previous systematic literature review studies in business settings barely specify journal continuity as the inclusion criteria of literature. Thus, this selection criterion is expected to be a novelty of this study by bringing new information to the SLR literature, thus contributing to the review studies in a business context.

Based on those explanations outlined earlier, the study objective of this research is to review scholarly work on fintech adoption using the SLR technique employing the Scopus database. The following research questions (RQs) are proposed in this study.

**RQ1:** What factors determine fintech service usage or adoption as available in academic literature indexed in Scopus?

**RQ2:** What categorizations can be made to group those factors stated in RQ1?

This paper is outlined as follows. This Section 1 is the introduction, providing the study's purpose, motivation, and knowledge contribution. The following section presents literature review, followed by materials and methods of the study. The Section 4 is results and discussion, followed by a conclusion in the final section.

## 2. Literature Review

This current study employs a systematic literature review (shortened as SLR) technique in reviewing the documents. SLR is a common review technique offering practitioners and researchers a categorized, organized perspective of the literature generated within a specific period [15]. The SLR approach of this research is based on the PRISMA framework adopted from the previous study [16].

SLR is a powerful tool to map and evaluate literature in any field. It has been used by researchers in diverse topics, such as accounting blockchain [17], fintech [15], working capital management [18], warehouse management [19], innovation implementation [20], open innovation [21], and digital leadership [22]. To perform an SLR study, a researcher can rely on a single database or various databases simultaneously, depending on the research purpose and scope. This study employs Scopus since it has extensive coverage [23] and is one of the major databases used in SLR studies by academia, providing complete information to be analyzed.

The SLR technique in this study is a theory-based review technique, one of the four common systematic review techniques, namely, domain-based, method-based, theory-based, and meta-analytical-based reviews [24]. Through this theory-based approach, the selected documents in this study are grouped based on the theoretical foundations, i.e., determining factors of fintech adoption among customers.

## 3. Materials and Methods

This study uses the Scopus database because it is one of the most trusted and reputable academic indexing bodies. Besides, Scopus has extensive coverage [23] and regular update for filtering non-reputable journals. For a journal to be indexed in Scopus, it must pass particular criteria, and it can later be delisted if it no longer fulfills particular criteria.

Literature research in this study was conducted using the following search string: TITLE (fintech AND adoption) AND (LIMIT-TO (SUBJAREA, "BUSI") OR LIMIT-TO (SUBJAREA, "ECON")) AND (LIMIT-TO (SRCTYPE, "j") OR LIMIT-TO (SRCTYPE, "p")). The search for documents was conducted on 19 September 2022. Figure 1 displays the process of selecting the documents for review in this study.

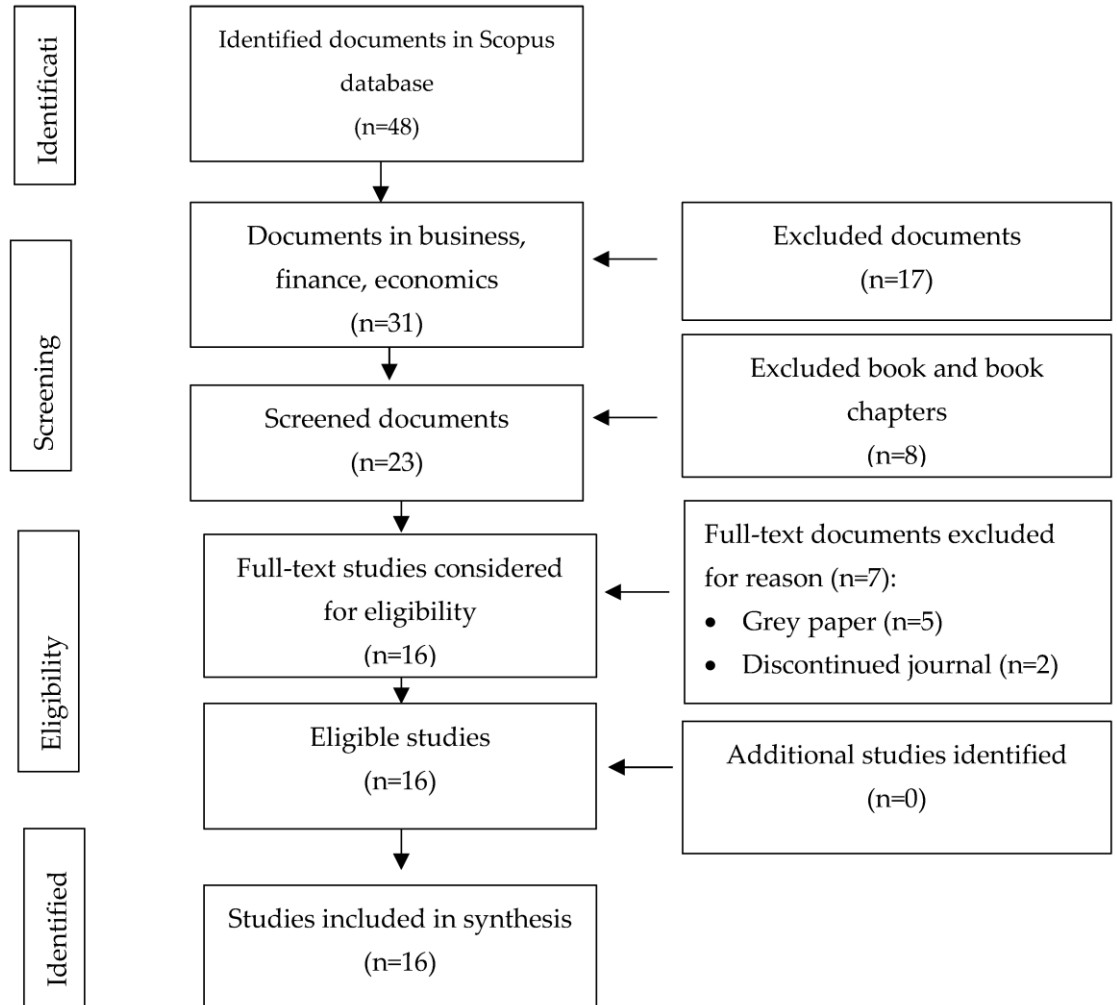

**Figure 1.** Document Selection.

To obtain the relevant literature on the factors determining fintech adoption or acceptance, a review of the content was performed on the sixteen final documents. Initially, we obtained 23 screened documents, meaning they relate to the topic we are studying. However, because of two reasons, we excluded seven papers in total. We found that five papers do not deal with determining factors of fintech adoption among customers, but bank adoption of fintech [25,26], fintech adoption for customer retention [27], adoption of artificial intelligence by both fintech and large companies [2], and fintech impact on financial inclusion across societies in different income levels [28]. Furthermore, we excluded two papers i.e., [29,30] because the journals publishing them are no longer indexed by Scopus. Delisting from Scopus may indicate that the journals are poor in terms of quality, or they can be categorized as predatory, thus must be of the concern of any individual or research intuition [31].

## 4. Results and Discussion

### 4.1. Publication Year

Documents in our study consist of journal articles published in the last four years, with the year 2019 as the oldest. This result indicates that the research paper about fintech adoption is relatively new because fintech itself experienced massive growth only in recent years. Furthermore, as indicated in Figure 2, the trend of fintech adoption research is increasing, with most documents (*n* = 7 or 43%) found in the year 2022. The number of studies in the year 2022 will undoubtedly be higher than shown in Figure 2 since the data collection in this study was performed in September 2022.

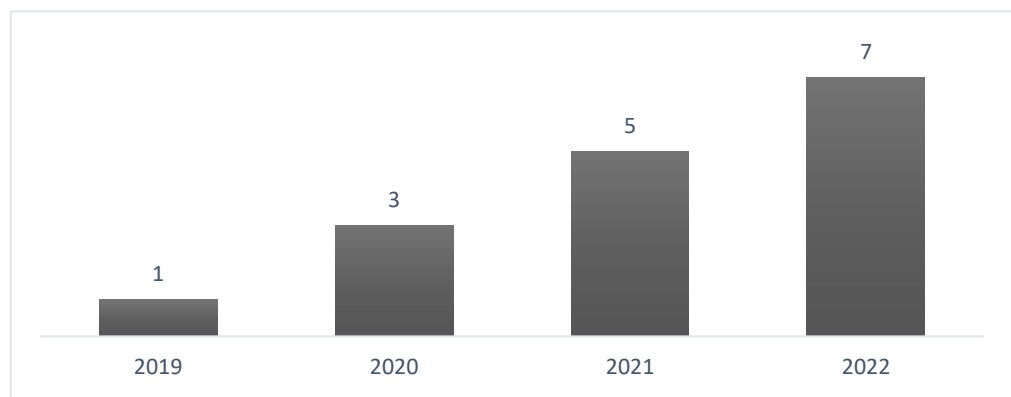

**Figure 2.** Publication year.

*4.2. Journal and Publisher*

The entire documents of this study are available in 16 different types of journals, meaning that there is not the most or least dominant slot. However, when seen from the publisher, most of the journals are published by Emerald Group Holdings Ltd., with eight papers in eight different journals (50%). These results may indicate that Emerald Group Holdings Ltd. has diverse publication slots and thus is considered more appropriate to select if one wishes to publish papers about fintech, primarily fintech adoption. The second position is held by Elsevier Inc. and MDPI AG, with two papers each. Considering the journal's best quartile of Scopus, ten out of 16 (62%) journals publishing the papers in our dataset are in the first quartile (Q1), indicating that the papers are of high quality. The journals and their publishers of the dataset in this study are shown in Table 1 as follows.

**Table 1.** Journals of Documents and Publishers.

| Journal Name | Best Scopus Quartile | Publisher |
|---|---|---|
| Journal of Financial Intermediation | Q1 | Academic Press Inc. |
| Journal of Business Research | Q1 | Elsevier Inc. |
| Finance Research Letters | Q1 | |
| African Journal of Economic and Management Studies | Q1 | Emerald Group Holdings Ltd. |
| Foresight | Q2 | |
| Industrial Management and Data Systems | Q1 | |
| International Journal of Bank Marketing | Q1 | |
| International Journal of Social Economics | Q2 | |
| Journal of Financial Regulation and Compliance | Q3 | |
| Journal of Islamic Marketing | Q2 | |
| Management Decision | Q1 | |
| Journal of Open Innovation: Technology, Market, and Complexity | Q1 | MDPI AG |
| Journal of Theoretical and Applied Electronic Commerce Research | Q2 | |
| Global Business Review | Q2 | Sage Publications India Pvt. Ltd. |
| Journal of Small Business Management | Q1 | Taylor and Francis Ltd. |
| E a M: Ekonomie a Management | Q1 | Technical University of Liberec |

*4.3. Selected Paper*

The dataset of this study consists of 16 relevant final documents published using different methods and used in different country contexts. Based on content analysis on the dataset, it was revealed that the entire papers (100%) are empirical papers using the quantitative method, meaning that they contain research findings. This study's dataset contains neither conceptual nor qualitative papers, indicating that fintech adoption has a sufficient theoretical background. For instance, TAM (technology acceptance model) and UTAUT (unified theory of acceptance and use of technology) are the most used theoretical foundation in the dataset of this study, in addition to the theory of planned behavior (TPB) and others. Furthermore, the sixteen documents in our dataset employ different country contexts in Asia, Europe, America, and Africa. It implies a global phenomenon of fintech adoption, both in developed and developing countries. Besides, research using various countries further diversifies the determinants of fintech adoption by capturing the country-specific characteristics.

In terms of determinants of fintech adoption, numerous factors significantly determine the adoption or intention to use fintech service. Various studies in our dataset present different types of determinants of this adoption. Besides, we found positive and negative significant determinants, with the most revealing the positive ones. The analysis results of fintech adoption are outlined in the subsequent sub-section.

*4.4. Determinants of Fintech Adoption*

Based on the SLR technique employed in this study, some theoretical and self-construct backgrounds have significantly affected fintech adoption among customers, as shown in Table 2. Based on that, this study groups those determinants into five clusters.

4.4.1. TAM-Related Determinants

For understanding the determinants of fintech adoption, most of the documents in our dataset are based on Technology Acceptance Model (TAM). TAM is the extension of the Theory of Reasoned Action (TRA). TAM was first put forth by Davis in 1989 to define the procedure through which people embrace and make use of new technology [32]. In our dataset, we found that Ref. [33] revealed consumers' attitudes, mass media, and subjective interpersonal norms as the factors which have a positive relationship with the adoption of robo-advisors (one of the fintech business models). Their finding is in line with Ref. [34], revealing that perceived ease of use and perceived usefulness affect mobile payment adoption among Dutch customers. Besides, the finding of Ref. [33] is supported by Refs. [35,36] on the effect of perceived usefulness and user attitude on fintech adoption. However, those findings are not supported by Ref. [37], finding that TAM-related factors do not affect fintech-service adoption in a bank-based financial system in Vietnam. In fact, they revealed latent customer needs for fintech service and customer knowledge (derived from innovation diffusion theory), which affect fintech adoption.

**Table 2.** Papers on fintech adoption.

| No | Authors | Title | Year | Method & Research Type | Theory | Country | Significant Independent Variables (Including Sign) |
|---|---|---|---|---|---|---|---|
| 1 | Ali et al. [38] | How perceived risk, benefit and trust determine user Fintech adoption: a new dimension for Islamic finance | 2021 | Quantitative & empirical | Theory of perceived risk (TPR), perceived benefit and trust | Pakistan | (+) User trust |
| 2 | Belanche et al. [33] | Artificial Intelligence in FinTech: understanding robo-advisors adoption among customers | 2019 | Quantitative & empirical | Technology acceptance model (TAM) | North America, Britain, Portugal | (+) Consumers' attitudes toward robo-advisors<br>(+) Mass media<br>(+) Interpersonal subjective norms |
| 3 | Chan et al. [39] | Towards an understanding of consumers' FinTech adoption: the case of Open Banking | 2022 | Quantitative & empirical | Unified theory of acceptance and use of technology (UTAUT) | Australia | (+) Performance expectancy<br>(+) Effort expectancy<br>(+) Social influence<br>(-) Perceived risk |
| 4 | Frederiks et al. [40] | The early bird catches the worm: The role of regulatory uncertainty in early adoption of blockchain's cryptocurrency by fintech ventures | 2022 | Quantitative & empirical | Resource-based view | Cross-countries | (+) Regulatory uncertainty has a positive effect on NTBFs' adoption of fintech (crypto) |
| 5 | Fu & Mishra [41] | Fintech in the time of COVID-19: Technological adoption during crises | 2022 | Quantitative & empirical | Not available | Cross-countries | (+) COVID-19 pandemic spread and lockdowns affect download finance app |
| 6 | Hasan et al. [34] | Evaluating Drivers of Fintech Adoption in The Netherlands | 2021 | Quantitative & empirical | TAM & VAM (value-based adoption model) | Netherland | (+) Perceived ease of use<br>(+) Perceived usefulness<br>(+) Safety<br>(+) Trust |
| 7 | Huarng & Yu [42] | Causal complexity analysis for fintech adoption at the country level | 2022 | Quantitative & empirical | Not available | Cross-countries | Combination of the followings:<br>(+) High values of innovation,<br>(+) Technology<br>(+) Entrepreneurship<br>(+) Economic development |
| 8 | Jünger & Mietzner [43] | Banking goes digital: The adoption of FinTech services by German households | 2020 | Quantitative & empirical | Not available (self-developed) | Germany | (+) Perceived trust<br>(+) Reliability<br>(+) Transparency requirement<br>(+) Financial literacy |
| 9 | Kakinuma [44] | Financial literacy and quality of life: a moderated mediation approach of fintech adoption and leisure | 2022 | Quantitative & empirical | Not available (self-developed) | Thailand | (+) Leisure<br>(+) Financial literacy + leisure |

**Table 2.** *Cont.*

| No | Authors | Title | Year | Method & Research Type | Theory | Country | Significant Independent Variables (Including Sign) |
|---|---|---|---|---|---|---|---|
| 10 | Mazambani & Mutambara [45] | Predicting FinTech innovation adoption in South Africa: the case of cryptocurrency | 2020 | Quantitative & empirical | Theory of Planned Behavior (TPB) | South Africa | (+) Attitude<br>(+) Perceived behavioral control |
| 11 | Ngo & Nguyen [37] | Consumer adoption intention toward FinTech services in a bank-based financial system in Vietnam | 2022 | Quantitative & empirical | TAM & innovation diffusion theory | Vietnam | (+) Customer latent needs for fintech service<br>(+) Customer knowledge |
| 12 | Rahim et al. [46] | Measurement and structural modelling on factors of Islamic Fintech adoption among millennials in Malaysia | 2022 | Quantitative & empirical | UTAUT | Malaysia | (+) Behavioral intention<br>(+) Facilitating conditions |
| 13 | Setiawan et al. [36] | User innovativeness and fintech adoption in Indonesia | 2021 | Quantitative & empirical | TAM, Institutional Theory (IT), and Individual Innovativeness Theory (IIT) | Indonesia | (+) Brand image<br>(+) Fintech perceived usefulness<br>(+) User attitude<br>(+) Financial literacy<br>(+) User innovativeness |
| 14 | Shubbangi Singh et al. [35] | What drives FinTech adoption? A multi-method evaluation using an adapted technology acceptance model | 2020 | Quantitative & empirical | TAM, UTAUT, ServPerfand & WebQual 4.0 | | (+) Perceived usefulness<br>(-) social influence |
| 15 | Solarz & Swacha-Lech [47] | Determinants of the adoption of innovative fintech services by millennials | 2021 | Quantitative & empirical | Not available (self-developed) | Poland | (+) H2: making decisions about choosing a financial institution based on the opinions about a financial institution in social media<br>(+) H4: Modernity applied solutions<br>(+) H7: Using a smartwatch is important<br>(-) H10: age<br>(+) H11: male, than female |
| 16 | Xie et al. [48] | Understanding fintech platform adoption: Impacts of perceived value and perceived risk | 2021 | Quantitative & empirical | UTAUT | China | (+) perceived value<br>(-) perceived risk<br>(+) social influence |

### 4.4.2. UTAUT-Related Determinants

The second-most documents in our dataset employ UTAUT as the basis of their research. UTAUT is developed from various studies which postulate that the determinants of any technology adoption or use are the four following constructs: performance expectancy, effort expectancy, social impact, and facilitating conditions [49]. The documents in our dataset report slightly different results. For instance, Ref. [39], studying the adoption of open banking in Australia, reported that performance expectancy, effort expectancy, and social influence are the three factors that positively determine open banking adoption. In comparison, Ref. [46] reports that behavioral intention and facilitating conditions are the factors that positively affect the adoption of Islamic Fintech services among millennials in Malaysia. Furthermore, in terms of social influence, Ref. [48] finds a positive effect on usage intention, while Ref. [35] reveals a negative result. It implies that adoption intention may or may not be explained by the customer environment, depending on internal and external factors.

### 4.4.3. TPB and TPR-Related Determinants

Our dataset only contains one paper using the Theory of Perceived Risk (TPR) and Theory of Planned Behavior (TPB) models as their theoretical foundation. In a Pakistani context, Ref. [38] uncovers that the customer's trust plays a significant role in adopting Islamic fintech services. The logic is that, trust is more essential than any other variable because customers deal with risk and uncertainty in their money. Thus, it is paramount for fintech firms to build customer trust and keep their promise. Once broken, customers may leave the company and switch to other fintech firms. On the other hand, using TPB as a research underpinning, Ref. [10] unveils that attitude and perceived behavioral control are two essential factors determining cryptocurrency adoption. Ref. [10] adds that subjective norm has a negative and insignificant role on the adoption intention due to customers' privacy and secrecy preference.

### 4.4.4. Other Theories

The study's dataset contains a single paper using other mainstream theories outlined earlier. Those papers employ the value-based adoption model (VAM), institutional theory, the resource-based view (RBV), and individual innovativeness theory (IIT). Using the VAM framework combined with TAM in the Dutch market, Ref. [34] uncovers that the VAM theory's variables, namely, perceived value earned and perceived value loss, did not exhibit any appreciable positive or negative relationships with fintech adoption. It could be a result of the Dutch mobile payment service providers not offering discounts or coupons to clients to entice them to utilize mobile payments. There is no loss of fees or associated costs while using mobile payments.

While in a study using a cross-country context and RBV theoretical lens for new technology-based firms (NTBFs), Ref. [40] reveals that regulatory uncertainty has a positive effect on NTBFs' adoption of fintech (crypto). The idea is that ambiguity can help NTBFs gather valuable resources and acquire a head start or competitive advantage. This result aligns with Ref. [36], integrating three theories (TAM, Institutional Theory, and Individual Innovativeness Theory), depicting that technology exposure plays a significant role in fintech adoption.

### 4.4.5. Self-Developed Constructs

In addition to some existing and popular theories or models affecting fintech adoption, our dataset also documents several variables that empirically affect the adoption. For instance, trust, not available in any popular models outlined above, is one factor that determines customers to select a fintech platform. Some studies confirm that trust is highly considered because the transaction in fintech is entirely online, without face-to-face interaction [34,38,43]. Thus, establishing customer trust should be in the fintech firm's

strategy to win the market competition, especially in today's highly competitive era where newcomers can quickly enter the market and the business landscape is prone to disruption.

Based on the review results and content analysis reported earlier, this study proposes a framework shown in Figure 3, which summarizes the determinants of fintech adoption in business and management literature. This framework consists of five theoretical foundations from which fintech adoption determinants are derived: TAM, UTAUT, TPB & TPR, other theories, and self-developed constructs. Besides, this framework answers the two research questions proposed earlier, for it outlines the variables or factors significantly affecting fintech adoption in the literature.

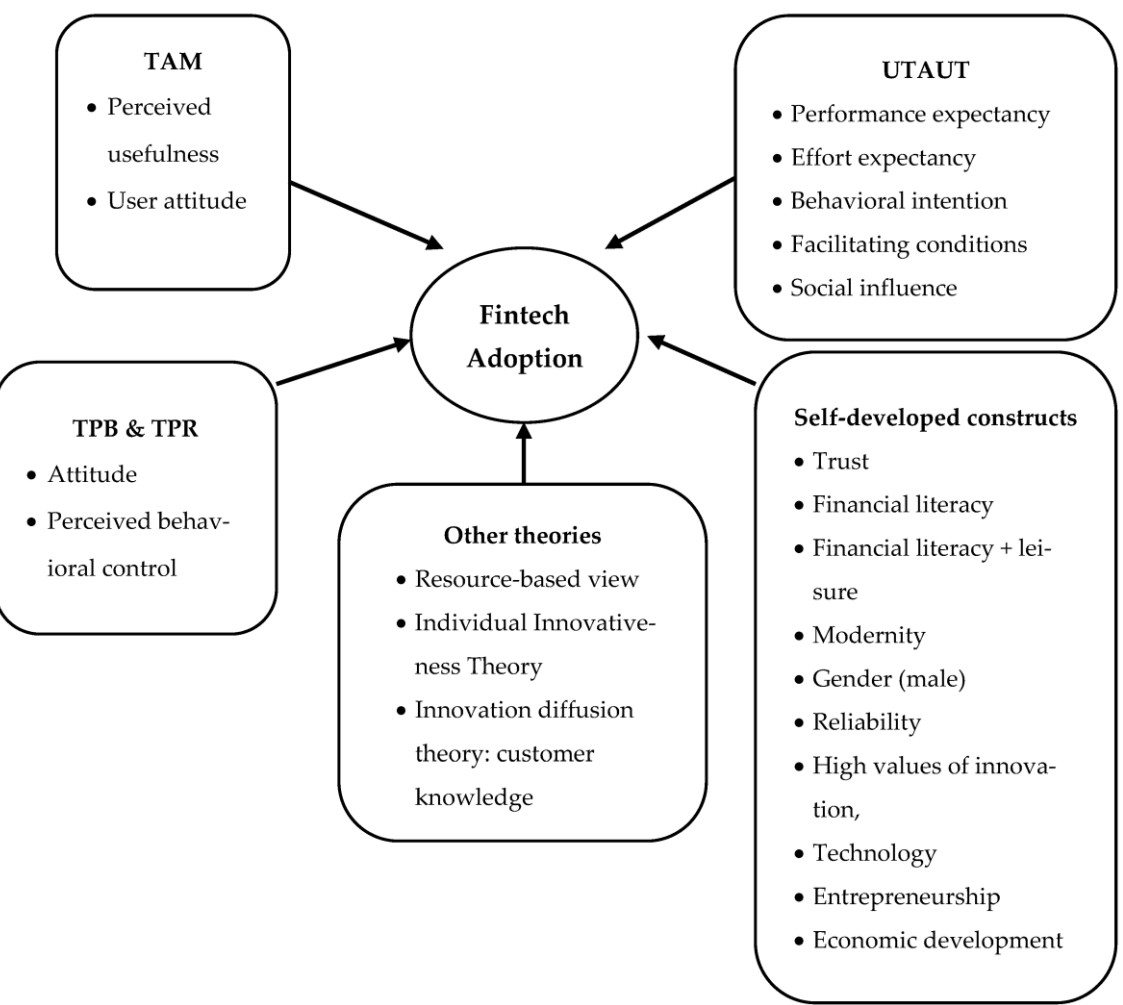

**Figure 3.** The proposed framework of fintech adoption determinants in literature.

## 5. Conclusions

This study employs the SLR technique to explore the factors determining fintech service usage or adoption as available in scholarly literature. The Scopus database was selected for data retrieval, and several inclusion criteria were applied to select the final documents to be reviewed. Subsequently, this study categorizes those factors obtained earlier into five clusters which become the theoretical foundations affecting fintech adoption among customers. The finding of this study reveals that various factors have been documented in the literature affecting fintech service adoption. Those factors in the literature are derived from various theoretical backgrounds, such as TAM, TPB, UTAUT, TPR, and other theories (such as RBV and institutional theory). From these theories, we found that TAM is the most-used theoretical background, followed by UTAUT. TAM is the most-selected theory

or model because it is the one that deals with the acceptance level of customers to the new technology.

Furthermore, this SLR-based study also reveals that some studies employed self-developed constructs and found various significant factors determining fintech adoption, such as trust and financial literacy. For instance, trust has been proposed as an essential element in fintech adoption because customers deal with fintech firms virtually, thus requiring high-level trust from the fintech platform. Acquiring customer trust is one of the keys to obtaining a long-term relationship with customers and is the key to a fintech firm's sustainability.

This SLR study contributes to the literature of SLR and fintech alike. First, this is among the first review study which presents the duality consisting of established theories and self-developed constructs of fintech adoption. This evidence shows that studies on fintech may keep developing in the future. Second, this study extends the method by employing journal continuity in Scopus indexation as one of the criteria for document inclusion in the SLR, which is rare in previous SLR studies. Third, this study presents the framework of fintech adoption determinants synthesized from the fintech literature, providing handy information for understanding the determining factors of fintech adoption.

Several implications can be drawn from the conclusions of this study. First, theoretically, this study reveals that fintech researchers use both the existing theories and self-developed constructs in explaining fintech adoption determinants, as shown in the proposed framework of this study. This fact encourages future fintech researchers to employ current theories and develop other constructs to contribute to fintech literature. Second, fintech companies can practically benefit from the proposed framework in this study to primarily maintain the loyalty of the customers. To do so, fintech managers or practitioners should consider both the theory-based and self-construct determinants, which might be more dynamic to change over time.

This study is not free from limitations. First, in terms of adoption, this study does not segregate the actual usage and intention to use due to the dissimilar theoretical backgrounds employed by documents in our study. Thus, further research may segregate those two types of adoption exposure to obtain more precise analysis results. Second, further studies may employ other databases, such as Web of Science or Google Scholar, to obtain analysis from different data sources. For example, due to its huge indexing coverage, Google Scholar will produce more results from different sources and journal levels. Thus, it will perhaps yield another result of fintech adoption.

**Author Contributions:** E.A.F. downloaded the dataset and prepared the manuscript. M.M., M.A. and M.H.A.B. interpreted the results and edited the manuscript before submission. All authors have read and agreed to the published version of the manuscript.

**Funding:** This research received no external funding.

**Institutional Review Board Statement:** Not applicable.

**Informed Consent Statement:** Not applicable.

**Data Availability Statement:** The dataset in this study is obtained from the Scopus database, which can be accessed by anyone having the Scopus subscription.

**Conflicts of Interest:** The authors declare no conflict of interest.

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
