# Peer review of "Factors Affecting Fintech Adoption: A Systematic Literature Review"

_fintech, doi:10.3390/fintech2010002_

Round 1

Reviewer 1 Report

1.     Introduction need to be extended. I don’t see many relevant concepts explained and still not clear for the readers. Also there should be purpose and objective of research well defined at the end of the section. It is important to highlight why this study is important and how it brings new information to the field of study. Author should provide enough references for the arguments presented in this section.

2.     Review of literature is missing and should be more focused on the past studies rather than proposing anything new. I don’t see many studies reviewed and referenced. It must be reviewed and extended by adding relevant and recent studies. 

Only 16 studies, why? There are many studies on Fintech adaptation. 

Baber, H. (2019). Relevance of e-SERVQUAL for determining the quality of FinTech services. International Journal of Electronic Finance9(4), 257-267.

Oladapo, I. A., Hamoudah, M. M., Alam, M. M., Olaopa, O. R., & Muda, R. (2021). Customers’ perceptions of FinTech adaptability in the Islamic banking sector: comparative study on Malaysia and Saudi Arabia. Journal of Modelling in Management.

Author Response

Dear reviewer,

Thank you so much for your constructive comments and feedback. Please find the attachment comprising the revised paper followed by our responses.

Notes: the red font in our manuscript is the additions or changes, yellow highlighted sentences are our responses.

Thank you

Reviewer 2 Report

Congrats on your paper!

A good paper that request minor revisions:

1. Do not include abbreviations before writing the whole words first; e.g in the abstract and keywords 

2. Elaborate more on the findings of your paper: what can we learn from this meta-analysis? what are possible theoretical and practical implications? 

Author Response

(The authors gave the same response as above.)

Author Response

(The authors gave the same response as above.)

Round 2

Reviewer 3 Report

Congratulations for the paper, the revision of the paper is in accordance with the comments and suggestions given